# Navigating Digital Transformation: Current Trends in Digital Competencies for Open Innovation in Organizations

**Lorena Espina-Romero** [1,*], **Doile Ríos Parra** [2], **José Gregorio Noroño-Sánchez** [3], **Gloria Rojas-Cangahuala** [1], **Luz Emerita Cervera Cajo** [4] and **Pedro Alfonso Velásquez-Tapullima** [5]

1   Escuela de Posgrado, Universidad San Ignacio de Loyola, Lima 15024, Peru; grojasc@usil.edu.pe
2   Centro de Investigaciones Sociales y Económicas, Universidad Popular del Cesar, Valledupar 200002, Colombia; doilerios@unicesar.edu.co
3   Facultad de Derecho y Ciencias Políticas, Universidad de Cartagena, Cartagena 130001, Colombia; jnoronos@unicartagena.edu.co
4   Escuela de Posgrado, Universidad César Vallejo, Lima 15314, Peru; lcerverac@ucv.edu.pe
5   Facultad de Ciencias Empresariales, Universidad Privada del Norte, Lima 15314, Peru; pedro.velasquez@upn.edu.pe
*   Correspondence: lespina@usil.edu.pe

**Abstract:** This study explored both the evolution and the integration of digital competencies within the context of open innovation, emphasizing the impact of technological advances across various sectors. The goal was to analyze documents indexed in Scopus from 2017 to 2023 using a quantitative and bibliometric approach. The methodology employed RStudio version 4.3.2 and Microsoft Excel 365 for data analysis, focusing on variables such as documents per year, h-index, total citations, and academic sources. The findings indicated a significant increase in research, highlighting a diversity of approaches, a trend towards international collaboration, and an evolution from basic concepts to complex applications, particularly the integration of digitization with sustainability and innovation. This research underscores the transformation of the business sector through digital competencies in open innovation and suggests further exploration into the effects of the pandemic on digital competencies, the role of artificial intelligence, the digital competencies–sustainable development relationship, and their application across different sectors.

**Keywords:** digital competencies; open innovation; technology; higher education; marketing; digital transformation

## 1. Introduction

Research on digital competencies today, especially in the context of open innovation, covers a wide and varied field [1–4]. This bibliometric study seeks to deeply explore this field, identifying trends, changes, and gaps in existing knowledge [5–9]. Digital competencies, defined as the skills and knowledge required to effectively navigate the digital environment, are critical in today's era of digital transformation, where technology and open innovation are cornerstones of almost every sector [10,11].

The bibliometric review method is suitable for addressing the research questions posed because of its ability to quantify the impact and trends of research in a specific field [12,13]. This method allows for a thorough analysis of the literature to identify trends, shifts in research focus, and underexplored areas [14–17]. Despite the growing literature on digital competencies in open innovation, there is a gap in how these competencies have evolved and how they are integrated in different contexts and sectors [18–24]. This gap warrants a detailed and at the same time updated study to serve as a basis for future research. Accordingly, this research poses five specific research questions (RQ1 to RQ5).

RQ1. What were the trends and changes in digital competencies research from 2017 to 2023 and how do these relate to open innovation in various areas?

RQ2. What are the trends and impact in the scientific literature on digital competencies in the context of open innovation, according to the number of publications, h-index, and total citations in relevant academic sources?

RQ3. What are the key trends and impacts of digital competencies on organizations, considering the world's most cited documents on open innovation?

RQ4. What are the status and trends in the production of knowledge on digital competencies in the context of open innovation worldwide, considering geographical distribution and academic impact?

RQ5. What are the future research directions to better understand digital competencies in the context of open innovation?

The main objective was to analyze the documents indexed in Scopus from 2017 to 2023 on the variables that were the subject of this study. This study will contribute a comprehensive analysis of the state of research on digital competencies within the framework of open innovation, highlighting key trends, impacts, and future directions. It will provide a solid foundation for researchers, guiding future research in the field.

This document will be structured as follows: after the introduction, a review of the literature will be conducted. This will be followed by the methodology, which provides a description of the methodological steps. Then, the results will be presented along with their respective analyses, addressing the research questions and interpreting the results. Finally, the conclusions will be presented, which include a synthesis of the findings as well as recommendations for future research and the implications and limitations of the study.

## 2. Trends in Digital Competencies and Open Innovation: A Review of the Literature

The literature review to support this study builds on several previous studies. An analysis of these studies is provided below, highlighting the main theories, approaches, and methodologies used, as well as the definitions of the variables under study according to the authors.

- Martínez-Bravo et al. [25] focus on digital competency, defining it as the safe and critical use of ICT for employment, learning, personal development, and participation in society. They also highlight the role of online communities based on the exchange of experiences, information, and knowledge. The study focuses on the virtual community ScolarTIC and reveals that its decentralized knowledge-sharing structure can aid in the development of digital competencies, with an emphasis on security and information.

- The study by Magni et al. [26] focuses on how digital competencies modify co-creation in higher education. It uses case study analysis, focusing on Little Genius International, an international school in English for digital natives. It emphasizes the importance of digital competencies in understanding student co-creation in higher education.

- Ibadango-Galeano et al. [27] focus on the competencies acquired by Educational Technology and Innovation teachers in the creation of digital stories within a telecollaboration project. Digital, narrative, creative, and didactic competencies are evaluated, indicating that the experience of creating digital stories is highly successful in teaching practice.

- Prendes-Espinosa et al. [28] address business and digital competencies as cross-cutting competencies in higher education. A model called EmDigital is developed, encompassing four areas, 15 sub-competencies, and 45 indicators related to digital business competency. This model has implications for assessment and training in higher education.

- As for Bartoli et al. [29], their study analyzes the impact of digitalization on product marketing with geographical indications (GIs) in agriculture. It highlights the need to enhance digital competencies for the integration of technologies in all business areas and for innovation in the GI market.

- Nuccio and Mogno [30] in their chapter review the literature on knowledge, skills, and competencies (KSCs) in the knowledge-based economy. The importance of KSCs in various fields is emphasized as well as the need for both creative and digital competencies in the workforce.

- Arnold et al. [31] focus on virtual collaborative learning–teaching and how immersive technologies develop competencies such as collaboration, virtual communication, and problem solving. The study is based on a formative assessment of skills in the Hotel Academy project and highlights the potential of virtual reality in higher education.
- Pintarić and Tomasović [32] focus on digital competencies related to open science and propose a framework that includes both generic and professional digital competencies. The need to enhance the digital competencies of future actors in open science is emphasized.

In general, these studies emphasize the importance of digital competencies in contexts such as higher education, teaching, marketing, and open science. However, there are gaps, such as further explorations into digital competencies in open innovation and a lack of focus on specific contexts or regions. These gaps can be addressed in the present study, which will contribute to a more comprehensive understanding of the literature in this field and identify emerging trends and research areas. Below is a comparative table (Table 1) that highlights why the present study is significant compared to the academic works cited in this section.

**Table 1.** Comparative table between studies.

| Aspect | Previous Studies | Present Study |
|---|---|---|
| Approach | Focus on digital competencies in specific contexts (education, marketing, open science, etc.). | Specific focus on digital competencies in the context of open innovation in various fields. |
| Research Questions | Various research questions depending on the focus of each study. | Five specific research questions that address the trends, impacts, and current state of digital competencies in relation to open innovation. |
| Temporal Scope | Several years of examined publications, but without a specific time focus. | Time-specific focus from 2017 to 2023, allowing for the identification of recent trends. |
| Relevance of Open Innovation | Focus on digital competencies without an explicit link to open innovation. | Highlights the relationship between digital competencies and open innovation, addressing how they are related in various areas. |
| Impact and Trends | Focus on describing digital competencies without an emphasis on impact and trends in the literature. | Assesses the impact and trends in the academic literature related to digital competencies in the context of open innovation. |
| Contribution | Contribute to knowledge in their specific areas. | Provides a comprehensive overview of the state of research on digital competencies in open innovation, identifying key trends and future directions. |
| Usefulness | Useful for researchers in specific fields (education, marketing, etc.). | Useful for researchers interested in understanding the relationship between digital competencies and open innovation in various fields, providing guidance for future activities. |

This study stands out for its specific focus on digital competencies in the context of open innovation as well as for its temporal approach, explicit relationship with open innovation, and ability to identify trends and future directions. This makes it a valuable contribution to the understanding of a significant research area today.

## 3. Materials and Methods

In this quantitative study with a bibliometric approach [12], five specific objectives were established to assess the evolution of trends in digital competencies research from 2017 to 2023 and identify key thematic changes and their impact on open innovation. The objectives were as follows:

O1. Evaluate the evolution of trends in digital competencies research from 2017 to 2023, identifying key thematic changes and their impact on open innovation.

O2. Evaluate the most relevant academic sources and their impact on the field of digital competencies in the context of open innovation.

O3. Synthesize key trends and impacts of digital competencies on organizations, based on an analysis of the most globally cited documents related to open innovation.

O4. Assess the production and impact of research on digital competencies in the context of open innovation worldwide, identifying the main contributing countries in terms of the number of documents and citations as well as the distribution of research by continent and the global gap in knowledge production.

O5. Explore emerging areas in digital competencies related to open innovation to guide future research and enhance practical application across various sectors.

### 3.1. Population and Sample

The population for this study consisted of a total of 18,658 publications related to digital competencies and open innovation from the period 2005 to 2023. The selected sample comprised 287 documents limited to the period 2017–2023. "Articles in Press" (Aip) were excluded to ensure data quality in the analysis (Figure 1).

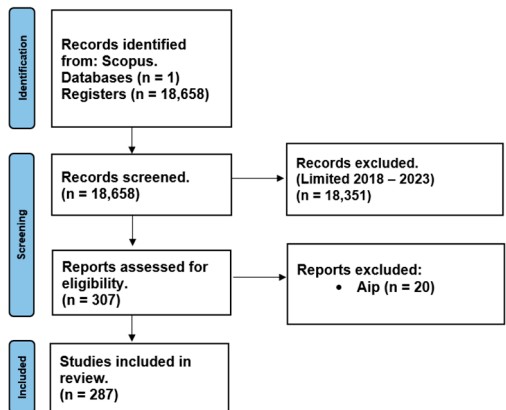

**Figure 1.** PRISMA method—process of sample selection.

### 3.2. Data Sources

A search was conducted in the Scopus database [33], using a combination of terms related to digital competencies and open innovation. The terms were as follows: "Digital competences" OR "Digital skills" OR "Digital literacy" OR "Digital competencies" OR "Digital capabilities" OR "Tech skills" OR "Technology literacy" OR "Computer literacy" OR "Information technology skills" OR "E-literacy" OR "Internet skills" AND "Open Innovation" OR "Collaborative Innovation" OR "External Innovation" OR "Crowdsourced Innovation" OR "Participative Innovation" OR "Networked Innovation" OR "Cooperative Innovation" OR "Distributed Innovation" OR "Collective Innovation" OR "Shared Innovation".

### 3.3. Variables Analyzed

Various variables of interest were collected and recorded from Scopus. The analyzed variables included the following:

- Documents per year of publication: Refers to the total number of documents published each year. This variable allows for the analysis of scientific productivity over time [34].
- h-index: Proposed by Hirsch in 2005, this variable measures the productivity and citational impact of an author's or journal's publications. An author has an h-index if h out of their Np articles has at least h citations each [35].
- Total citations: This is the total sum of times a work by an author, article, or source has been cited. It reflects the impact or influence of the published research [36].
- Sources (journals or conferences): Refers to the scientific journals or conferences in which the analyzed documents have been published. This variable helps identify knowledge dissemination platforms in a specific field [36].

- Annual growth rate (%): Measures the percentage of annual growth or decline in scientific production in a specific field. It offers insight into the evolution of the study area [12].
- Average age of documents: Calculates the average age of the analyzed documents, thus providing insight into the temporal relevance of research in a specific field [37].
- Average citations per document: This indicator shows the average number of citations received by each document, providing a measure of the average impact of works in a specific field [38].
- Cited references: Identifies whether bibliographic references are cited in the analyzed documents. This dataset allows for the study of the theoretical and methodological foundations upon which current research is built [39].
- Author keywords: These are keywords provided by the authors of the documents. These are useful for identifying the main research topics and thematic trends within a field of study [40].
- Most influential authors, journals, documents, and countries: Identifies authors, journals, documents, and countries with the greatest impact in a field of study, based on measures such as the number of citations and visibility [41].
- Percentage of international co-authorship: Measures the percentage of documents resulting from international collaborations. This indicator shows global scientific cooperation [42].

These variables were crucial for assessing the quantitative and qualitative aspects of publications in the field of digital competencies and open innovation.

### 3.4. Methods of Analysis

For the analysis of the data collected in this study, a combination of tools and analytical techniques was employed to address the objectives of this research. Below are the methodologies applied using RStudio and Microsoft Excel 365.

### 3.4.1. Use of RStudio

RStudio, an Integrated Development Environment (IDE) for the R programming language, was utilized as a tool for statistical analysis and data generation to populate the tables. The selection of RStudio was due to its ability to handle large datasets and its flexibility for complex statistical analyses [13]. Figure 2 depicts the state of Bibliographic Metadata Integrity according to the report from this software.

In this study, the following methods were applied using RStudio:

- Network approach for thematic analysis (Objective 1): A network analysis model was implemented to identify thematic trends over time. This approach involved the creation of nodes for key themes and the evaluation of their interconnectedness based on co-occurrence in the literature. The igraph package in R was used to build and analyze the network, allowing for the identification of thematic clusters and their evolution.
- Generation of thematic maps and bar charts (Objective 1): Using RStudio's graphics functionality, thematic maps were created to represent the distribution of themes over the years. Furthermore, for an interpretation of trends over time, bar charts were employed to visualize the number of publications per year.

| Metadata | Description | Missing Counts | Missing % | Status |
|---|---|---|---|---|
| AB | Abstract | 0 | 0.00 | Excellent |
| AU | Author | 0 | 0.00 | Excellent |
| DT | Document Type | 0 | 0.00 | Excellent |
| SO | Journal | 0 | 0.00 | Excellent |
| LA | Language | 0 | 0.00 | Excellent |
| PY | Publication Year | 0 | 0.00 | Excellent |
| TI | Title | 0 | 0.00 | Excellent |
| TC | Total Citation | 0 | 0.00 | Excellent |
| C1 | Affiliation | 2 | 0.70 | Good |
| CR | Cited References | 3 | 1.05 | Good |
| DI | DOI | 21 | 7.32 | Good |
| DE | Keywords | 26 | 9.06 | Good |
| RP | Corresponding Author | 52 | 18.12 | Acceptable |
| ID | Keywords Plus | 182 | 63.41 | Critical |
| NR | Number of Cited References | 287 | 100.00 | Completely missing |
| WC | Science Categories | 287 | 100.00 | Completely missing |

**Figure 2.** Integrity of bibliographic metadata (source: RStudio).

### 3.4.2. Use of Microsoft Excel 365

For data organization, analysis, and visualization, Microsoft Excel 365 was employed. Excel was selected for its intuitive interface and data handling capabilities for complementary analyses, such as the below:

- Analysis of academic sources (Objective 2) and most cited documents (Objective 3): Filters and sorting functions in Excel were applied to classify and analyze academic sources and documents according to their relevance and number of citations. This process enabled the identification of the most significant contributions to the field of study.
- Assessment of the global distribution of publications (Objective 4): Excel was used to map the geographical distribution of publications and their impact on citations, thus facilitating the visualization of global and regional patterns in research.
- Identification of emerging trends and creation of a future research agenda (Objective 5): Through the analysis of data generated and visualized in Excel, emerging trends were identified to propose a future research agenda. This analysis was supported by the synthesis of the findings from this study.

### 3.4.3. Integration of Tools and Techniques

The integration of RStudio and Microsoft Excel allowed for a diverse approach to data analysis. While RStudio facilitated complex analyses, Excel was used for more direct analysis and visualization tasks. This combination ensured both the depth and accessibility of analysis, enabling an exploration of data from multiple angles.

### 3.5. Limitations

This study faced inherent limitations due to its methodological approach and scope. For instance, selecting a single database, Scopus, and the exclusive use of RStudio may have limited the breadth of the reviewed literature, omitting relevant studies on other platforms. Additionally, by excluding press articles, emerging research may have been overlooked. On the other hand, technological evolution and changes in the dynamics of open innovation can make findings less representative over time. The generalization of the results is also limited by the geographical concentration of the analyzed studies, which may not reflect variations in digital competencies and open innovation in different contexts.

External factors, such as governmental policies and global crises, not deeply examined, also play a role in how these competencies and innovations are developed and implemented.

### 3.6. Ethics and Legal Considerations

We utilized a variety of software, search engines, and translators incorporating artificial intelligence (AI) technologies to enhance various aspects of our research: Microsoft Word for grammar and style suggestions, Microsoft Excel for data analysis and visualization suggestions, DeepL for accurate translations, ChatGPT for comparing translations, and Google Search for optimizing information retrieval. However, it is important to note that these resources did not replace data interpretation or scientific conclusions extraction. Ethical and legal principles were respected in the collection and analysis of bibliometric data.

## 4. Results and Discussion

### 4.1. Main Information on the Selected Data

The main information was obtained from the overview generated by RStudio. Figure 3 shows an annual growth of 59.56% from 2017 to 2023, indicating an expansion and interest in the field of digital competencies related to open innovation. With 210 sources and 287 documents, this field exhibits a diverse knowledge base, suggesting interdisciplinary interest and a broad application of digital competencies.

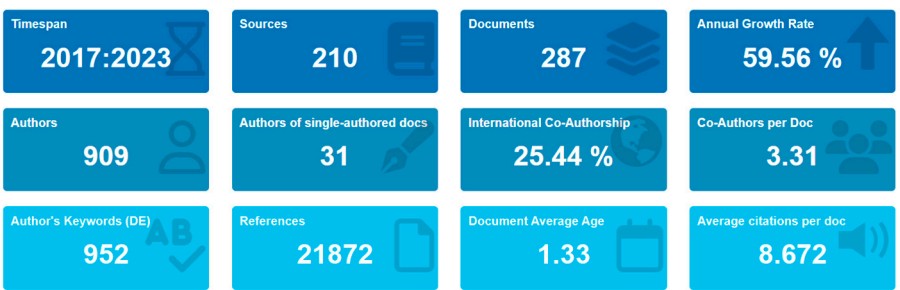

**Figure 3.** Main information (source: RStudio).

The average age of the documents was 1.33 years, which was relatively low. This suggests that most of the research in this field is recent. An average of 8.672 citations per document was a significant indicator of the relevance and impact of research in this field. With 952 author keywords (DE) and 909 authors, including 31 authors of single-authored documents, a variety of research approaches and interests within the variables under study were demonstrated. The 25.44% international co-authorship highlighted global collaboration in this field, although there is room for increased international collaboration.

Most of the documents were articles (196), followed by conference papers (47), book chapters (27), review documents (14), and books (3). This indicates a preference for publication in journals and conferences, which is common in fields oriented towards applied research.

In summary, this analysis shows that digital competencies in the context of open innovation are growing, with a strong focus on the recent literature and diversity of approaches and contributions, with a trend towards international collaboration, although there is still room for growth in this latter aspect.

### 4.2. Analysis of Progression in Digital Competencies Research: Key Trends and Thematic Changes from 2017 to 2023 in the Context of Open Innovation

To comply with Objective 1, this section evaluates the evolution of trends in digital competencies research from 2017 to 2023 to identify thematic changes and their impact on open innovation. This section examines the main themes and the quantity of documents produced in the selected period, allowing us to identify how research interests and focus areas in the field of digital competencies have evolved. The themes for each year were obtained using RStudio software version 4.3.2 through the analysis of the Conceptual

Structure and its network-focused approach known as the Thematic Map. This approach enabled the identification of trends and relationships among key concepts over the years. Figure 4 shows a significant increase in the number of studies, rising from 6 documents in 2017 to 99 in 2023, indicating an interest in research on the variables under study.

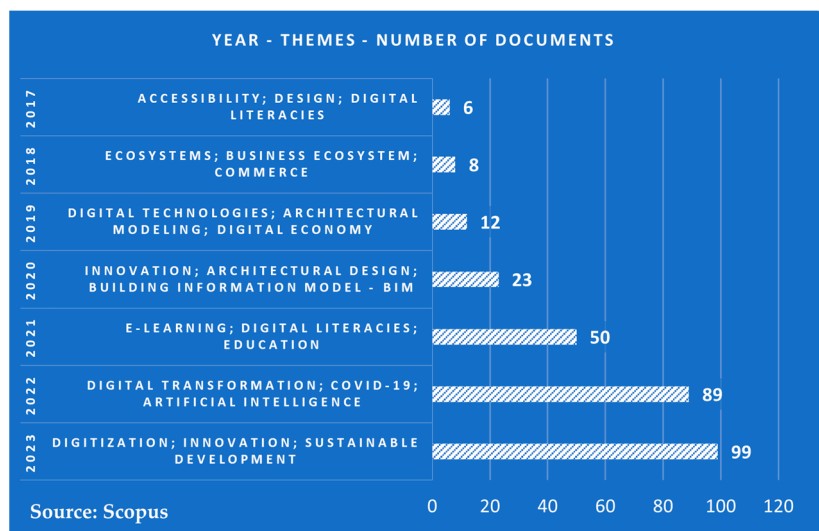

**Figure 4.** Evolution of digital competencies: 2017–2023.

In the years 2017–2018, the themes were centered on aspects such as 'ecosystems', 'accessibility', 'design'', and 'digital literacies', suggesting an exploration phase to establish foundations in the field [25,43,44]. In 2019–2020, there was a shift towards more technical topics such as 'digital technologies', 'architectural modeling', 'digital economy', and 'Building Information Modeling (BIM)'. This reveals a transition towards the application of digital competencies in more specialized areas [45–49].

In 2021, with themes like 'e-learning', 'digital literacies', and 'education', the focus seemed to return to fundamentals, possibly influenced by the pandemic and the need for digital competencies in education [4,28,50]. In 2022, themes such as 'digital transformation', 'COVID-19', and 'artificial intelligence' indicated a response to global challenges, with a focus on advanced technologies [1,51,52]. In 2023, themes like 'digitization', 'innovation', and 'sustainable development' suggested an expansion of the field, integrating digitization with global objectives such as sustainability [53–55].

Over the years, research themes have evolved from basic concepts to more complex applications and, finally, to an integration of digitization with topics like sustainability and innovation. The emergence of themes like 'COVID-19', 'artificial intelligence', and 'sustainable development' demonstrates how digital competencies are responding to global trends. Martínez-Bravo et al.'s and Ibadango-Galeano et al.'s studies align with the initial phases and the return to educational fundamentals, highlighting the importance of online communities and the creation of digital stories in teaching [25,27]. In turn, Magni et al. and Prendes-Espinosa et al. consider the transition towards technical applications and the focus on digital competencies in higher education and digital business [26,28]. Additionally, Bartoli et al., Nuccio and Mogno, Arnold et al., and Pintarić and Tomasović align with the integration–expansion phase in 2023, focusing on digitalization for sustainability, innovation, and the importance of digital competencies in fields such as agriculture, the economy, and higher education, among others [29–32].

In summary, research on digital competencies has evolved to not only address technical and educational aspects but also to respond to global challenges and contribute to open innovation in sectors such as technology, education, and sustainable development.

### 4.3. Analysis of the Most Relevant Sources

This section evaluates the 10 most relevant academic sources and their impact on the field of digital competencies in the context of open innovation to meet Objective 2. These sources were identified using RStudio. The number of documents, h-index, and total citations (TC) in Table 2 are analyzed.

**Table 2.** Relevant sources.

| Relevant Sources | Articles | h-Index | TC |
|---|---|---|---|
| "Sustainability (Switzerland)" | 26 | 7 | 146 |
| "Journal of Open Innovation: Technology, Market, and Complexity" | 7 | 4 | 293 |
| "Communications in Computer and Information Science" | 6 | 1 | 2 |
| "Journal of Business Research" | 5 | 3 | 83 |
| "Technological Forecasting and Social Change" | 5 | 2 | 48 |
| "ACM International Conference Proceeding Series" | 4 | 1 | 1 |
| "Frontiers in Psychology" | 4 | 1 | 6 |
| "Lecture Notes in Computer Science (Including Subseries Lecture Notes in Artificial Intelligence and Lecture Notes in Bioinformatics)" | 4 | 1 | 11 |
| "Cogent Business and Management" | 3 | 2 | 6 |
| "Heliyon" | 3 | 2 | 7 |

This study reveals some trends based on the provided data. Each source is analyzed below:

- Sustainability (Switzerland): This journal led in the number of articles (26) and had a respectable h-index (7), with a total of 146 citations. This suggests that it is a primary source in the field of digital competencies and open innovation, with a significant impact on the academic community [56–58].
- Journal of Open Innovation: Technology, Market, and Complexity: Despite having only 7 articles, its h-index (4) and total citations (293) were impressive. This indicates that although the number of publications is lower, the influence and quality of the work are very high, reflecting a considerable impact on open innovation research [4,53–55,59].
- Communications in Computer and Information Science: With 6 articles but a low h-index (1) and only 2 citations, it seems that this source is less influential in the field, with a lesser impact on the research community [60,61].
- Journal of Business Research: This journal had 5 articles with an h-index of 3 and 83 citations. This indicates a notable contribution to the study of digital competencies, being a relevant source of information and analysis in this field [62–65].
- Technological Forecasting and Social Change: Similarly, with 5 articles but a lower h-index (2) and 48 citations, this suggests that this source plays an important role, although perhaps not as central as others in digital competencies research [20,66–68].
- ACM International Conference Proceeding Series and Frontiers in Psychology: Both with 4 articles but a low h-index (1) and very few citations (1 and 6, respectively), this could indicate that they are emerging or niche sources in the field of open innovation and digital competencies [69–72].
- Lecture Notes in Computer Science (including subseries): With 4 articles, an h-index of 1, and 11 citations, this shows a modest contribution to the field [73,74].
- Cogent Business and Management and Heliyon: With 3 articles each, an h-index of 2, and total citations of 6 and 7, respectively, these are sources that contribute to the field but with limited impact compared to the main sources [75–80].

In summary, this analysis highlights the dominance of journals like "Sustainability" and "Journal of Open Innovation" in digital competencies research within the framework of open innovation. Although there are other sources with significant contributions, these two appear to be the most influential in terms of the number of publications and citations, demonstrating their relevance and authority in the field.

*4.4. Analysis of the Most Cited Documents in Digital Competencies and Open Innovation*

In this section, the trends and impacts of digital competencies in organizations were summarized according to the analysis of the most cited documents in the framework of open innovation, a crucial issue in the era of digital transformation, to comply with Objective 3. The information included in Table 3 was generated with RStudio.

Analyzing the data from Table 3, several key trends in the most cited global documents stand out.

- A focus on digital transformation during the COVID-19 pandemic: The study by Priyono et al. [59] was the most cited, highlighting digital transformation in SMEs during the pandemic and emphasizing digital adaptation in times of crisis.
- Digital competencies and organizational performance: Khin and Ho [81] and Heredia et al. [82] address the relationship between digital technology, digital competencies, and organizational performance. This indicates an interest in how digitalization directly impacts business success.
- Challenges and opportunities of Industry 4.0: Queiroz et al.'s work on Industry 4.0 and digital supply chain capabilities highlights the challenges of digitalization in the industrial sector [83].
- Gap in digital marketing competencies: Herhausen et al. [84] identify a gap in digital marketing competencies, signaling a critical development area for companies in the digital era.
- The role of personnel in digital transformation: Cetindamar et al. [85] and El Hilali et al. [45] focus on the role of employees in digital transformation, addressing the topics of digital literacy and sustainability.
- A microfoundational perspective in the era of digital transformation: Scuotto et al. [62] and Kamberidou [47] explore how SMEs and women entrepreneurs adapt to the digital economy, showing an interest in how individuals and small organizations handle digital transformation.
- Adoption of AI technologies in manufacturing: Kinkel et al.'s study on the adoption of artificial intelligence in manufacturing reveals an interest in the application of advanced technologies in traditional sectors [86].

**Table 3.** Most cited documents.

| Author | Title | Total Citations |
|---|---|---|
| Priyono et al. (2020) [59] | "Identifying Digital Transformation Paths in the Business Model of SMEs during the COVID-19 Pandemic" | 238 |
| Khin and Ho (2019) [81] | "Digital technology, digital capability and organizational performance: A mediating role of digital innovation" | 234 |
| Queiroz et al. (2019) [83] | "Industry 4.0 and digital supply chain capabilities: A framework for understanding digitalization challenges and opportunities" | 143 |
| Herhausen et al. (2020) [84] | "The digital marketing capabilities gap" | 96 |
| Heredia et al. (2022) [82] | "How do digital capabilities affect firm performance? The mediating role of technological capabilities in the new normal" | 70 |
| Cetindamar et al. (2021) [85] | "Understanding the role of employees in digital transformation: conceptualization of digital literacy of employees as a multi-dimensional organizational affordance" | 70 |
| El Hilali (2020) [45] | "Reaching sustainability during a digital transformation: a PLS approach" | 70 |
| Scuotto et al. (2021) [62] | "A microfoundational perspective on SMEs' growth in the digital transformation era" | 55 |
| Kamberidou (2020) [47] | "Distinguished women entrepreneurs in the digital economy and the multitasking whirlpool" | 55 |
| Kinkel et al. (2022) [86] | "Prerequisites for the adoption of AI technologies in manufacturing—Evidence from a worldwide sample of manufacturing companies" | 46 |

These trends demonstrate an interest in how digital competencies and open innovation are reshaping the business sector, emphasizing digital adaptation, the importance of human capital in digital transformation, and the integration of AI in traditional sectors.

### 4.5. Analysis of Trends, Impacts, and Geographic Distribution

In this section, the research in the field of study was globally assessed, highlighting the leading countries in production and impact, continental distribution, and global disparities in knowledge generation. To fulfill Objective 4, this research revealed several trends based on the data provided by RStudio, which are visualized in Figure 5. Here is an analysis of these trends:

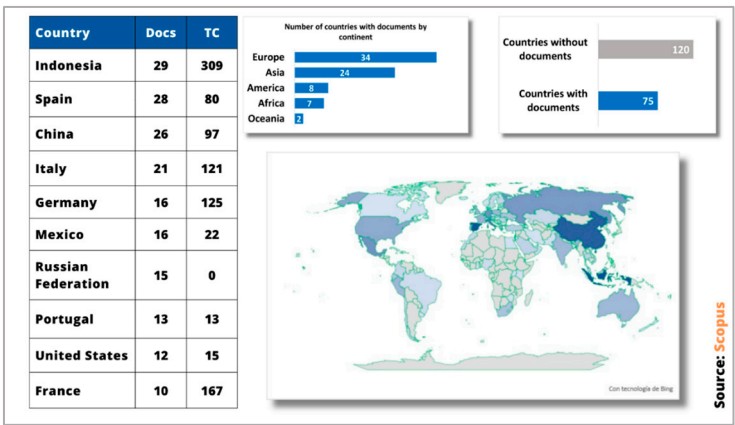

**Figure 5.** Global map of research in digital competencies and open innovation.

#### 4.5.1. Key Trends of the Ten Countries with the Highest Number of Documents and Their Total Citations

Indonesia led in the number of papers (29), with a high total of citations (309), suggesting a significant impact on research on the variables under study [87–90]. Spain followed closely behind with 28 documents, although its total citations were relatively low (80), indicating less influence in the field compared to Indonesia [54,91,92]. China, with 26 papers and 97 total citations, shows a steady engagement in research, albeit with a moderate impact in terms of citations [93–95].

Italy and Germany had a similar number of papers (21 and 16, respectively), but Germany had a higher citation total (125 versus Italy's 121), suggesting a profound influence on the academic community [29,96–98]. Mexico and the Russian Federation demonstrate commitment in the field (16 and 15 documents, respectively), but with varying impact, as Mexico had 22 citations and Russia had none [70,99–101]. Portugal, the United States, and France round out the list, with France standing out due to a high number of citations (167), despite having fewer documents (10), indicating highly influential research [69,102–105].

#### 4.5.2. Number of Countries with Documents by Continent

Asia and Europe led in terms of continental participation, with 24 and 34 countries respectively involved in research, representing 46.67% and 65.22% of their countries. America and Africa had moderate participation, with 8 and 7 countries, respectively (17.14% and 12.96% of their countries). Oceania showed the lowest participation with only 2 countries involved, representing 20% of the countries on the continent.

#### 4.5.3. Total Global Countries with and without Documents

There were 75 countries (38.46%) that have contributed documents in this area, indicating a global scope of research. However, there are still a large number of countries (120, or 61.54%) that have not contributed to this area, indicating possible knowledge gaps or a lack of resources for research in these countries [106]. In short, this analysis reveals global participation in research on digital competencies within the context of open innovation,

with variations in impact and influence among countries. This information can be crucial for understanding how different regions contribute to and benefit from open innovation in the digital field.

### 4.6. Future Research Agenda: Digital Competencies in the Context of Open Innovation

To fulfill Objective 5, this document presents a future research agenda focused on exploring emerging areas related to digital competencies within the framework of open innovation. This agenda is based on trends from previous analyses in this study, proposing research questions that address everything from the impact of the pandemic on digital transformation to the evolution of digital competencies in the context of emerging technologies and collaborations between countries (see Table 4).

**Table 4.** Key areas and questions for future research.

| Approach | Research Question |
|---|---|
| Post-pandemic digital transformation | How has the pandemic impacted the acceleration of digital competencies in different sectors and what lessons can we learn? [107,108] |
| Artificial intelligence and digital competencies | How is artificial intelligence redesigning digital competencies in the labor market and education? [109–111] |
| Digital competencies for sustainable development | How can digital competencies be integrated to promote sustainable development and what roles can they play in the UN Sustainable Development Goals? [112,113] |
| Emergence of new technologies and education | Which digital competencies will be considered essential in response to technologies such as augmented reality, robotics, and blockchain? [88,114] |
| Open innovation and organizational transformation | How can organizations leverage open innovation to implement digital competencies effectively? [56,85] |
| Emerging trends in digital competencies | What new trends in digital competencies have emerged post-2023 and how are they expected to evolve? [8,49] |
| Digital competencies in various sectors | How does the effectiveness of digital competencies implementation vary across different industries? [115,116] |
| Digital competencies and leadership | What digital competencies are crucial for leaders looking to drive open innovation and digital transformation? [117,118] |
| Digital literacy gaps | What are the most significant gaps in digital literacy between demographic groups and how can they be addressed? [119,120] |
| Challenges and opportunities of Industry 4.0 | What are the challenges and opportunities presented by Industry 4.0 and how can companies prepare? [121,122] |
| Influencing factors in research | What factors drive the development of research on digital competencies in different regions? [74,123] |
| Gaps and inequalities in international research | Why do some countries show a high volume of publications with a low number of citations and what does this reveal? [124,125] |
| International collaborations and their impact | How do transnational collaborations influence the production of research in digital competencies? [126,127] |
| Practical application of research | How does research in digital competencies translate into effective open innovation practices? [128,129] |
| Strategies to increase research participation | What strategies can countries with less research adopt to increase their participation? [130,131] |

This agenda provides a comprehensive perspective on the issues that shape the field of digital competencies in the context of open innovation. It is hoped that addressing these questions will not only advance the understanding of these issues but also positively influence the actions of global organizations and governments. Adapting to technological changes and the demand for digital competencies represents a challenge but also an opportunity to foster inclusive and sustainable development in the digital age.

*4.7. Synthesis of Research Results (2017–2023)*

Below is a summary table of the main results aligned to each of the research questions (RQ1 to RQ5) of the document (Table 5).

**Table 5.** Synthesis of results.

| Research Question | Main Results |
|---|---|
| RQ1—Trends and changes in digital competencies research (2017–2023) | Annual growth of 59.56% in research from 2017 to 2023. Evolution of topics from 'ecosystems' and 'digital literacies' to 'digital technologies', 'digital economy', and the integration of digitalization with sustainability and innovation. |
| RQ2—Trends and impact in the scientific literature | Publications and citations in journals such as "Sustainability (Switzerland)" and "Journal of Open Innovation". Influential articles on digital transformation, digital technology and organizational performance, and Industry 4.0 challenges. |
| RQ3—Key trends and impacts of digital competencies in organizations | Influence of digital competencies on digital adaptation during crises (e.g., COVID-19). Role of employees in digital transformation. |
| RQ4—Status and trends in knowledge production | Leadership of Indonesia and Spain in number of documents. Global distribution shows significant differences in research production and impact. |
| RQ5—Future research directions | Proposed areas: impact of the pandemic on digital competencies, influence of AI on the labor market, and digital competencies for sustainable development. |

This table summarizes how digital competencies are evolving and the importance of digital research to meet future challenges.

**5. Conclusions**

This research has shown significant growth in the field of digital competencies for open innovation, demonstrating an evolution in the adoption of new technologies and methodologies. The bibliometric analysis applied in this study highlights an expansion of study topics, ranging from the integration of artificial intelligence to the importance of sustainability in open innovation.

The results show an annual increase of 59.56% in scientific production on the variables under study, underscoring the importance of digital competencies in the innovation domain. Emerging themes identified include digital transformation, artificial intelligence, and their impact on sustainability, indicating a shift towards more integrative and sustainable approaches in open innovation. Additionally, influential sources were identified, with journals such as "Sustainability (Switzerland)" and "Journal of Open Innovation: Technology, Market, and Complexity" being highlighted. The analysis of global research output and impact emphasized the contributions of countries such as Indonesia and Spain while also noting knowledge gaps elsewhere.

The findings suggest that policymakers should consider strategies to foster the development of digital competencies, promote open innovation, and facilitate the integration of new technologies to maintain competitiveness and sustainability in a globally digitized market.

Exploring the impact of the pandemic on digital competencies, the role of artificial intelligence in open innovation, and the relationship between digital competencies and sustainable development is recommended. These areas are relevant for future research that can contribute to an understanding of the evolution of open innovation in the digital era. Additionally, further exploration of knowledge and resource gaps between countries is suggested to promote more inclusive and collaborative research in the field of digital competencies and open innovation.

The findings are essential for researchers, professionals, and policymakers interested in the development of digital competencies and the promotion of open innovation. The results highlight the urgency of a comprehensive approach that integrates technology, sustainability, and global collaboration to address the challenges of digital transformation.

It is important to acknowledge that this study has limitations, such as the use of the Scopus database, which may not have encompassed all relevant research sources.

Additionally, specific search criteria could have influenced the results. Furthermore, the bibliometric approach has inherent limitations in its ability to capture research quality.

**Author Contributions:** Conceptualization, L.E.-R. and J.G.N.-S.; methodology, D.R.P. and G.R.-C.; software, L.E.C.C.; validation, L.E.-R., P.A.V.-T. and D.R.P.; formal analysis, P.A.V.-T.; investigation, L.E.-R.; resources, D.R.P.; data curation, J.G.N.-S.; writing—original draft preparation, L.E.-R.; writing—review and editing, J.G.N.-S.; visualization, L.E.-R. and G.R.-C.; supervision, P.A.V.-T. and G.R.-C.; project administration, L.E.C.C. All authors have read and agreed to the published version of the manuscript.

**Funding:** This research received no external funding.

**Institutional Review Board Statement:** Not applicable.

**Informed Consent Statement:** Not applicable.

**Data Availability Statement:** Not applicable. This study has a bibliometric approach and the data used were generated from the Scopus database.

**Acknowledgments:** We thank the developers of the AI tools that were essential in our study: Microsoft Word and Excel, for improving texts and data analysis; DeepL, for its precise translations; ChatGPT, for facilitating the comparison of text translations; and Google Search, for optimizing information search. Their contribution to our research was applied always respecting ethical and legal principles.

**Conflicts of Interest:** The authors declare no conflicts of interest.

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
