# Peer review of "Navigating Digital Transformation: Current Trends in Digital Competencies for Open Innovation in Organizations"

_sustainability, doi:10.3390/su16052119_

Round 1
Reviewer 1 Report
Comments and Suggestions for Authors
Dear Authors, Congratulations on the research. Very interesting insights.
The research is well-structured, logical and well-developed. I have no comments to add to improve it. Maybe letter (size) can be replaced in some of the presented tables.
Author Response
"Please see the attachment."

Reviewer 2 Report
Comments and Suggestions for Authors
The article treats a very actual subject.
More limitations should be added at subchapter 3.5. Conclusions state more limitations.
Some citations are strangely introduced into the text, the text seems to rather contain explanations given by the authors than a cited information. Examples: [34], [12], [48-50], [4,45–47,51], [52,53], [54-57], [20,58-60], ..., [29,88–90], etc. Please make sure that citations are appropriate.
It is useful to compare the obtained results with similar analysis from the literature. If any unexpected or contradictory results emerged during the study, the Discussion section provides possible explanations or hypotheses to account for them. Also, this section discusses the broader implications of the findings and their relevance to the field or to practical applications.
Author Response
"Please see the attachment."

Reviewer 3 Report
Comments and Suggestions for Authors
Please refer to the appended file.
A list of strengths is provided.
The list of defects is formulated.
Suggestions for improvements are given.
It is a moderately interesting exercise.

Comments on the Quality of English LanguageMinor editing of English language required
Author Response
"Please see the attachment."

Round 2
Reviewer 2 Report
Comments and Suggestions for Authors
The article can be published in the actual format.